# RELATION-AUGMENTED DIFFUSION FOR LAYOUT-TO-IMAGE GENERATION

## ABSTRACT

Existing layout-to-image generation methods often struggle in complex scenes with multiple objects, frequently exhibiting issues such as missing objects, positional errors, and semantic inconsistencies. These shortcomings largely stem from a fundamental inability to model inter-object relationships, which limits their capacity to capture spatial and relational cues effectively. To address these challenges, we propose *Relation-Augmented Diffusion*, a novel framework for layout-to-image generation that explicitly models inter-object relations and implicitly coordinates background-object interactions. We introduce a relation bounding box computation module to spatially encode object interactions, transforming abstract relations into concrete visual representations. These are further embedded into a topological scene graph via a graph convolutional network, enabling bidirectional reasoning between objects and their relations. Additionally, we employ a layout fusion module to harmonize implicit background-object spatial dependencies, which integrates global layout structures with background features to enhance overall scene coherence. Extensive experiments on HICO-DET, COCO-Position, and T2I-CompBench demonstrate that our framework significantly outperforms state-of-the-art methods in generating spatially and semantically consistent images. The code will be available at GitHub/XXX.

## 1 INTRODUCTION

Text-to-image generation models (Li et al., 2023; Nichol et al., 2021; Rombach et al., 2022; Saharia et al., 2022b) have recently achieved remarkable progress, enabling the generation of high-quality and diverse images based on textual prompts. Recent text-conditioned autoregressive and diffusion models, such as DALL·E (Ramesh et al., 2021; 2022a), Imagen (Saharia et al., 2022b), and Stable Diffusion (Rombach et al., 2022), have demonstrated the ability to produce images with high fidelity. Many works have extensively explored the use of class labels (Dhariwal & Nichol, 2021; Zheng et al., 2022), text (Radford et al., 2021; Ramesh et al., 2022b; Saharia et al., 2022c), images (Saharia et al., 2022a; Karras et al., 2021; Zhang et al., 2023b), and other modalities (Johnson et al., 2018; Farshad et al., 2023) to guide the direction of image generation. However, vanilla models like Stable Diffusion struggle to correctly and comprehensively design prompts when generating complex images containing multiple objects. Even with carefully crafted text prompts, text-guided models (Rombach et al., 2022; Nichol et al., 2021; Saharia et al., 2022b) still exhibit issues such as missing objects, incorrect positioning, and misclassified shapes. This is primarily due to the inherent ambiguity of text and the limited ability of current models to effectively capture and represent spatial information.

Utilizing spatial conditions to control the placement of objects in images has always been a focused area (Park et al., 2019; Zhang et al., 2023a; Li et al., 2024). In recent years, Layout-to-Image (L2I) generation models (Li et al., 2021; Zhao et al., 2019; Zhou et al., 2024; Xie et al., 2023; Hoe et al., 2024; Zheng et al., 2023; Cheng et al., 2024) have provided fine-grained controllability for image synthesis tasks by explicitly specifying object categories and geometric information such as positions. However, most existing methods simplify layouts as collections of independent instances (Zhou et al., 2024; Cheng et al., 2024), focusing on the reproduction of spatial attributes for individual objects while neglecting the semantic modeling of interactive relations between objects. This limitation becomes particularly pronounced in multi-instance complex scenes, they frequently exhibit typical issues like spatial-logical mismatches and functional-semantic disconnections, leading to reduced scene plausibility (Hoe et al., 2024). Although existing works (Chen et al., 2024; Li et al., 2023; Mao

et al., 2023) have attempted to mitigate these problems, they fundamentally lack the capability for structured representation of interactive relations.

To address these issues, we propose a *Relation-Augmented Diffusion* framework that enhances layout-to-image generation by explicitly modeling inter-object relations and implicitly harmonizing background-object interactions. Unlike existing approaches that treat objects as isolated entities, our method introduces a Relation Bounding Box Computation Module to dynamically derive explicit spatial representations of object interactions. These relation bounding boxes are computed based on the geometric distribution of objects within triples (e.g., relative positions, overlapping regions), transforming abstract textual relations into concrete visual regions. This explicit inter-object relation modeling ensures that interaction semantics are spatially grounded and directly supervised during generation. Moreover, we construct a dynamic topological graph using a Graph Convolutional Network (GCN) (Kipf & Welling, 2016), which establishes a bidirectional information flow between relation bounding boxes and entity objects, enabling the model to dynamically harmonize the consistency of object attributes and interaction semantics. Finally, for implicit background-object coordination, we employ a Layout Fusion Module, inspired by Zheng et al. (2023), which integrates global layout structures with background features to ensure that background regions interact harmoniously with foreground objects, addressing the limitations of prior works that treat the background as a negative space.

To evaluate our method's capability in generating images under layout conditions and its semantic generation ability, we conduct extensive experiments on recognized benchmarks: the HICO-DET dataset (Chao et al., 2018), COCO-Position (Zhou et al., 2024; Lin et al., 2014), and T2I-CompBench (Huang et al., 2023). The experimental results demonstrate that our proposed method achieves the best performance across multiple benchmarks. The significant improvement highlights our framework's ability to model fine-grained interactions that are challenging for existing baseline models. Additionally, the results on COCO-Position (Zhou et al., 2024; Ma et al., 2023) and T2I-CompBench (Huang et al., 2023) also confirm that our method maintains the strong spatial generation capabilities of L2I generation models. Our main contributions are summarized as follows:

- We introduce the Relation-Augmented Diffusion framework, which abstracts inter-object relations as independent structural entities for dedicated processing, significantly improving semantic-spatial consistency in generated outputs.
- Our framework uniquely addresses both explicit inter-object relations—via a novel relation bounding box computation module—and implicit background-object dependencies through a layout fusion module that harmonizes foreground objects with their surrounding context.
- Our method achieves significant improvements over baseline approaches across multiple benchmarks. It not only establishes a new relation-centric paradigm for complex scene generation but also opens innovative directions for designing multimodal conditional diffusion models.

## 2 RELATED WORK

### 2.1 TEXT-TO-IMAGE GENERATION

Early conditional GANs (Reed et al., 2016; Zhang et al., 2021) laid the foundation for controllable image synthesis by conditioning generation. Recently, they have been largely superseded by diffusion and autoregressive approaches—e.g., DALL-E (Ramesh et al., 2021; 2022a), Imagen (Saharia et al., 2022b), Stable Diffusion (Rombach et al., 2022)—which fuse text (via CLIP/T5 encoders (Radford et al., 2021; Raffel et al., 2020)) with UNet (Ronneberger et al., 2015) and cross-attention (Vaswani et al., 2017) for high-fidelity outputs. However, purely text-driven methods struggle to control spatial layout without restrictive priors or extra fine-tuning (Nichol et al., 2021; Li et al., 2023; Ramesh et al., 2022a).

### 2.2 LAYOUT-TO-IMAGE GENERATION

Layout-to-Image (L2I) generation has evolved from early GAN-based frameworks (Sun & Wu, 2019; 2021), which faced limitations in output quality. Recent works (Chen et al., 2024; Mao et al., 2023; Zheng et al., 2023; Xie et al., 2023) adapt pre-trained text-to-image diffusion models to incorporate

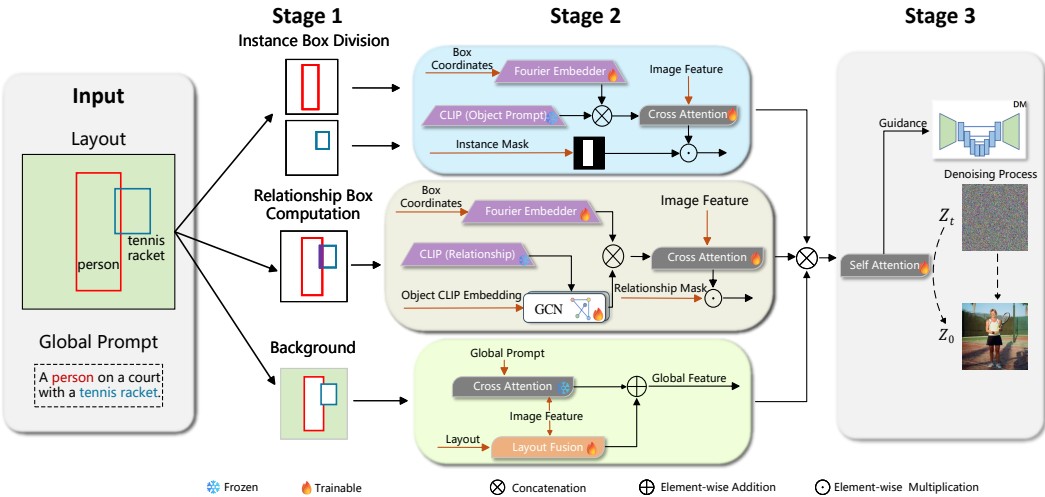

Figure 1: The proposed relation-augmented diffusion framework. The pipeline begins with partitioning the input layout into individual object bounding boxes, followed by the computation of relation bounding boxes to capture explicit interactions between objects (*Stage 1*). A GCN dynamically models bidirectional information flow between objects and relations (*brown, Stage 2*), while the layout fusion module harmonizes background-object dependencies (*green, Stage 2*).

bounding boxes or semantic maps through fine-tuning or attention modulation (Yang et al., 2023). Concurrently, synergies with large language models (LLMs) (Feng et al., 2023) demonstrate how layout planning and instruction-following can be delegated to text-driven agents.

## 2.3 RELATION MODELING

Explicit modeling of inter-object relations has been extensively studied across a range of computer-vision and multimodal tasks. In visual relation detection, Lu et al. (2016) introduced a joint object–predicate model that leverages language priors to predict triplets. Building on this, Krishna et al. (2017) released the Visual Genome dataset and demonstrated scene-graph generation by annotating dense graphs of objects and their pairwise relations. Outside of generation, relational reasoning modules—such as Relation Network (Raposo et al., 2017)—aggregate pairwise feature interactions to improve performance on visual question answering and few-shot learning.

## 3 METHOD

### 3.1 OVERVIEW

**Problem Definition.** In our work, the input consists of a global text prompt $T$ and a layout $L = \{\mathbf{b}^1, \mathbf{b}^2, ..., \mathbf{b}^N\}$ that includes all objects, where each bounding box $\mathbf{b}^i = [x_1^i, y_1^i, x_2^i, y_2^i]$ corresponds to a class name description $\mathbf{d}^i$ in the object set $D = \{\mathbf{d}^1, \mathbf{d}^2, ..., \mathbf{d}^N\}$. Given the input of text prompt and layout, our task is to generate an image $I$ using a diffusion model that not only respects the spatial constraints of the layout but also explicitly incorporates the relations among objects.

**Method Overview.** Our approach follows a clear and unified pipeline that emphasizes two main innovations: explicit inter-object relation modeling and implicit background-object spatial integration. As shown in Fig. 1, we first partition the overall layout into individual object bounding boxes. Building upon this, we propose a novel Relation Bounding Box Computation Module to derive explicit relation bounding boxes from pairs of object boxes. These relation boxes are designed to capture salient, explicit interactions. While object features are extracted using established cross-attention mechanisms, our method introduces a novel processing pipeline for relation features. Specifically, we integrate a trainable Graph Convolutional Network (GCN) to fuse objects cues, ensuring that relational information is semantically enriched and spatially coherent. Finally, to manage the implicit spatial dependencies between the background and objects, a dedicated Layout Fusion Module is employed to blend the global layout with background features.

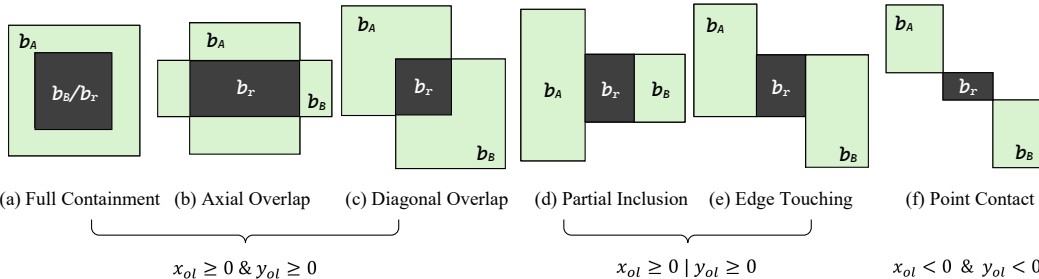

(a) Full Containment    (b) Axial Overlap    (c) Diagonal Overlap    (d) Partial Inclusion    (e) Edge Touching    (f) Point Contact

$$x_{ol} \geq 0 \,\&\, y_{ol} \geq 0 \qquad\qquad x_{ol} \geq 0 \,|\, y_{ol} \geq 0 \qquad x_{ol} < 0 \,\&\, y_{ol} < 0$$

Figure 2: Visualization of the Relation Bounding Box Computation Module. The module dynamically derives relation bounding boxes from pairs of object bounding boxes, adapting to their spatial configurations. We have listed various possible spatial relations between two objects, and our relation calculation module computes based on these relations to obtain abstract relational bounding boxes.

## 3.2 RELATION MODELING

**Object Bounding Box Division.** In previous work (Zhou et al., 2024; Xie et al., 2023; Hoe et al., 2024) for multi-instance generation tasks, multiple objects are often handled as separate subtasks, such as through hierarchical sampling. A corresponding mask $\mathbf{M}^i$ is generated for each box, where pixels inside the box are set to 1 and others to 0. The objective is to extract features $\mathbf{O}^i$ for each object such that:

$$\mathbf{O}^i = \arg\min_{\mathbf{O}^i}(\|\mathbf{O}^i - \mathbf{O}^{real}\|\mathbf{M}^i), \tag{1}$$

where $\mathbf{O}^{real}$ represents an objectively existing correct feature. The above formulation lays the foundation for our subsequent inter-object relation modeling.

**Relation Bounding Box Computation.** Previous layout-based generation approaches often consider each object as an independent entity, overlooking the influence of object relations on semantic consistency. In our work, the core novelty is the explicit modeling of inter-object relations.

Inspired by Egenhofer & Herring (1991), our method explicitly models inter-object relations by computing a relation bounding box $\mathbf{b}_r = [x_1^r, y_1^r, x_2^r, y_2^r]$ for each object pair. As shown in Fig. 2, the computation of $\mathbf{b}_r$ adapts to different spatial configurations, ensuring that it accurately reflects the area where the interaction occurs. Our approach classifies them based on the projection relationships of the objects along the x-axis and y-axis. Let the bounding boxes of objects A and B be denoted as $\mathbf{b}_A = [x_1^A, y_1^A, x_2^A, y_2^A]$ and $\mathbf{b}_B = [x_1^B, y_1^B, x_2^B, y_2^B]$. We primarily consider two aspects: overlap and containment. Define the degree of overlap between two objects along the x-axis and y-axis as $x_{ol}$ and $y_{ol}$:

$$x_{ol} = \min(x_2^A, x_2^B) - \max(x_1^A, x_1^B), \qquad y_{ol} = \min(y_2^A, y_2^B) - \max(y_1^A, y_1^B). \tag{2}$$

Each axis is considered *overlap* if $x_{ol}$ or $y_{ol} \geq 0$, otherwise *separated*. If $x_{ol} \geq 0$, the projections overlap o the x-axis; if $x_{ol} < 0$, the projections are separated on the x-axis, and similarly for the y-axis. If objects have an overlap relationship, their spatial relationship is further refined based on the projections on the two axes. When two objects overlap along one or both axes, the containment relationships of their projections onto each axis should be examined further; this analysis characterizes the precise spatial interactions between their bounding boxes. The projection onto the x-axis and y-axis is denoted as $\mathbf{x}_p$ and $\mathbf{y}_p$, using the signs of $x_{ol}, y_{ol}$ and axis-wise containment, the pairwise spatial relation is classified into six types (Fig. 2), a more detailed classification process is provided in the appendix. This relational bounding box computation module transforms implicit contextual cues into explicit structural inputs that guide the subsequent image generation process.

## 3.3 EXPLICIT INTER-OBJECT RELATION GROUNDING

**Object Feature Extraction.** For general objects, we first pre-process the text label and bounding box into an intermediate representation. In particular, we use the pre-trained CLIP (Radford et al., 2021) text encoder to encode the text of objects as a representative text embedding and use Fourier embedding to encode their respective bounding boxes following GLIGEN (Li et al., 2023). The fused object feature is given by:

$$\mathbf{h}_o^i = [f_{text}(d_i), \text{MLP}(\text{Fourier}(\mathbf{b_i}))] \tag{3}$$

where [·] represents the concatenation, and $f_{text}$ represents the pre-trained text encoder of CLIP (Radford et al., 2021). In the Stable Diffusion pipeline, its UNet inputs image features and text description into the cross-attention layer to obtain the residual, and then adds it to the image features to determine generated content. A trainable Cross-Attention layer then integrates these object features with the image features during the diffusion process:

$$\mathbf{O}^i = \text{Softmax}\left(\frac{\mathbf{Q}\mathbf{K}^{i^T}}{\sqrt{d}}\right)\mathbf{V}^i \cdot \mathbf{M}^i, \tag{4}$$

where $\mathbf{Q}$ is obtained from the image feature map, $\mathbf{K}^i$ and $\mathbf{V}^i$ are obtained from the grounded phrase token $\mathbf{h}_o^i$ in Eq. (3). The Instance Mask $M^i$ ensures precise spatial localization during training, guaranteeing that objects are generated in the correct regions.

**Relation Feature Extraction.** Previous works (Farshad et al., 2023) in text-to-image generation have treated relation predicates as categorical tokens to be fed into the CLIP (Radford et al., 2021) text encoder for obtaining relation embeddings, an approach our method also employs. However, we consider that relations in image generation differ from objects: while objects can be treated as individual entities, relationships are inherently tied to and constrained by their subject and object. Therefore, in our work, unlike the processing of instances, relations are modeled with a dedicated pipeline. We employ a trainable Graph Convolutional Network (Kipf & Welling, 2016) (GCN) to model objects and relations as triple structures. By using the GCN to perform information propagation and aggregation on the relation nodes, we obtain relation features that incorporate object information. Concretely, given input vectors $\mathbf{v}_s$, $\mathbf{v}_o$, $\mathbf{v}_r$ for subject, object and relation embedding, our goal is to obtain relation vectors that incorporate information from both the subject and object entities. Typically, a GCN employs three functions, $g_s$, $g_o$, and $g_r$, to compute the output vectors. To obtain the relation features, we utilize only the $g_r$ function to derive the relation vectors, and we also use the Fourier Embedder to obtain the spatial features. Our trainable GCN then propagates information and aggregates these cues via:

$$\mathbf{v}_r' = g_p(\mathbf{v}_i, \mathbf{v}_r, \mathbf{v}_j), \tag{5}$$

which is further enriched with spatial features through Fourier embedding:

$$\mathbf{h}_r^i = [\mathbf{v}_r', \text{MLP}(\text{Fourier}(\mathbf{b_r^i}))]. \tag{6}$$

After independently processing the relation features, we follow the same procedure as with the object features by using a trainable cross-attention layer to interact with the image features:

$$\mathbf{R}^i = \text{Softmax}\left(\frac{\mathbf{Q}\mathbf{K}_r^{i^T}}{\sqrt{d}}\right)\mathbf{V}_r^i \cdot \mathbf{M}_r^i, \tag{7}$$

where $\mathbf{K}_r^i$ and $\mathbf{V}_r^i$ are obtained from the grounded relation phrase token $\mathbf{h}_r^i$ in Eq. (6). Similarly, the relation mask here is derived from the mentioned relation bounding box computation module, ensuring that the interactions between objects are accurately localized to the correct regions. By multiplying the attention output with $\mathbf{M}_r^i$, our method confines the relation features strictly within the pre-determined spatial region, which not only reinforces the explicit interaction cues from the relation bounding box but also mitigates the risk of spatial ambiguity. In complex scenes where object interactions are subtle or overlapping, the relation mask acts as a precise spatial guide, ensuring that the relational information is incorporated only within the intended regions.

### 3.4 Implicit Background-Object Harmonizing

After obtaining the features of all instances and relations, the generation process within the bounding boxes annotated by the Layout is guided. Unlike earlier works that only use a global text prompt to generate background features, our method acknowledges that the background plays an implicit role in defining the scene. In our multi-object generation task, the background template helps the model distinguish between different regions, ensuring that the content in each region is independent and accurate. Illustrated in Fig. 1, we utilize global prompt $T$ to obtain the background feature $\mathbf{h}_{bg}$ in a manner similar to Eq. (6), with the background mask $\mathbf{M}_{bg}$, in which positions containing the instance are assigned a value of 0, while all other positions are marked as 1.

Previous generation works often use background masks and global textual prompts as background features to control the generation process. However, this approach fails to provide independent

Table 1: Quantitative results of different methods on HICO-DET benchmark. We report the HOI Detection Score based on the FGAHOI (Ma et al., 2023) protocol under both Default and Known Object settings. The table presents the test results of the FGAHOI detector using Swin-Tiny and Swin-Large backbones.

| Method | Swin-Tiny (mAP) | | | | Swin-Large (mAP) | | | |
| | Default | | Known | | Default | | Known | |
| | Full | Rare | Full | Rare | Full | Rare | Full | Rare |
|---|---|---|---|---|---|---|---|---|
| Stable Diffusion (Rombach et al., 2022) | 0.65 | 0.68 | 0.66 | 0.70 | 0.64 | 0.83 | 0.65 | 0.84 |
| GLIGEN (Li et al., 2023) | 21.73 | 15.35 | 23.31 | 17.24 | 23.99 | 19.56 | 24.89 | 20.37 |
| InteractDiffusion (Hoe et al., 2024) | 29.53 | 23.02 | 30.09 | 24.93 | 31.56 | 26.09 | 32.52 | 27.04 |
| BoxDiff (Xie et al., 2023) | 19.25 | 16.83 | 18.43 | 15.39 | 21.76 | 12.89 | 20.21 | 12.56 |
| MIGC (Zhou et al., 2024) | 27.53 | 25.01 | 26.65 | 25.27 | 28.18 | 21.34 | 25.73 | 22.33 |
| HiCo (Cheng et al., 2024) | 30.28 | 25.98 | 28.37 | 25.64 | 29.17 | 23.48 | 26.59 | 24.03 |
| DreamRenderer (Zhou et al., 2025) | 29.80 | 25.32 | 29.51 | 26.63 | 30.25 | 24.92 | 28.14 | 26.49 |
| **Ours** | **32.14** | **26.54** | **31.49** | **27.90** | **32.28** | **26.21** | **33.14** | **30.56** |

control signals for each instance, making it difficult for the model to distinguish between the features and relations of different instances. As shown in Fig. 1, inspired by Zheng et al. (2023), we employ a Layout Fusion Module to integrate the layout with the global text prompt. Combining the layout with the background improves the model's scene understanding and background–instance interactions. We fuse the layout result with the cross-attention output via element-wise addition to produce background guidance for generation:

$$\mathbf{h}_{bg} = [f_{text}(T), \text{MLP}(\text{Fourier}(\mathbf{b}_{bg}))] \tag{8}$$

$$\mathbf{O}_{bg} = \text{Softmax}\left(\frac{\mathbf{Q}\mathbf{K}_{bg}^T}{\sqrt{d}}\right)\mathbf{V}_{bg} \cdot \mathbf{M}_{bg} + \text{LF}(L, f_{image}(I)), \tag{9}$$

where $b_{bg}$ can be regarded as the coordinates of regions outside all object boxes, $\mathbf{Q}$ is derived from the image feature for the diffusion, and $\mathbf{K}_{bg}$ and $\mathbf{V}_{bg}$ are obtained from $\mathbf{h}_{bg}$. $\text{LF}(\cdot)$ represents the layout fusion process, and $f_{image}$ denotes the CLIP image encoder.

### 3.5 LAYOUT-CONDITIONAL DIFFUSION MODEL

**Combine Results.** To summarize, in all the above operations, we can get the sequences of objects, relations and background, i.e., $\mathbf{O}_N = \{\mathbf{O}^1, ..., \mathbf{O}^{N_1}, \mathbf{R}^1, ..., \mathbf{R}^{N_2}, \mathbf{O}_{bg}\} \in \mathbb{R}^{(N_1+N_2+1, C, H, W)}$ and $\mathbf{M}_N = \{\mathbf{M}^1, ..., \mathbf{M}^{N_1}, \mathbf{M}_r^1, ..., \mathbf{M}_r^{N_2}, \mathbf{M}_{bg}\} \in \mathbb{R}^{(N_1+N_2+1, 1, H, W)}$, where $N_1$ represents the number of instances, and $N_2$ represents the number of relations. After obtaining the above sequence, we process it using Self-Attention as guidance for the diffusion model. It enables extensive interaction between object features, thereby enhancing the model's expressive capability and enabling it to handle more complex generation tasks such as multi-instance generation.

**Training Loss.** We use the original denoising loss proposed in DDPM (Ho et al., 2020) and Stable Diffusion (Rombach et al., 2022) as our main training loss function:

$$\min_{\theta'} \mathcal{L}_{\text{LDM}} := \mathbb{E}_{z, \epsilon \sim \mathcal{N}(0, I), t}\left[\|\epsilon - \epsilon_{\theta, \theta'}(\mathbf{z}_t, t, T, L, D)\|_2^2\right], \tag{10}$$

where $\theta$ represents the frozen parameters of the pre-trained stable diffusion, and $\theta'$ represents the trainable parameters in our proposed framework.

## 4 EXPERIMENTS

We train and evaluate models at $512 \times 512$ resolution. We initialize our model with the pre-trained StableDiffusion v1.5 (Rombach et al., 2022). For the COCO benchmark, we use COCO 2017 (Lin et al., 2014) to train our model. To get the descriptions of instances, we use stanza (Qi et al., 2020) to split the global text prompt. During training, the original parameters of the base model remain frozen, while only the trainable modules marked in the figure are trained. We use AdamW (Kingma & Ba, 2017) optimizer with a constant learning rate of $1e^{-4}$, and train the model for 100 epochs with batch size 8 on four NVIDIA GeForce RTX 4090 GPUs. For inference, we employ diffusion sampling steps of 50 with the EulerDiscreteScheduler (Karras et al., 2022) sampler.

Table 2: Quantitative results of different methods on COCO-Position.

| Method | FID | mIoU | AP | AP50 | AP75 | CLIP Score |
|---|---|---|---|---|---|---|
| Stable Diffusion (Rombach et al., 2022) | **23.56** | 21.60 | 0.80 | 2.71 | 0.42 | **25.69** |
| BoxDiff (Xie et al., 2023) | 25.15 | 33.28 | 3.29 | 12.27 | 1.08 | 23.79 |
| Layout Diffusion (Zheng et al., 2023) | 25.94 | 57.49 | 23.45 | 48.10 | 20.70 | 18.28 |
| GLIGEN (Li et al., 2023) | 26.80 | 71.61 | 40.68 | 68.26 | 42.85 | 24.61 |
| MIGC (Zhou et al., 2024) | 24.52 | 77.38 | 54.69 | 84.17 | 61.71 | 24.66 |
| HiCo (Cheng et al., 2024) | 23.87 | 75.31 | 57.22 | 80.04 | 63.29 | 25.22 |
| DreamRenderer (Zhou et al., 2025) | 24.34 | 78.04 | 56.51 | **85.45** | 62.58 | 24.40 |
| Ours | 25.09 | **79.21** | **59.67** | 84.93 | **64.73** | 24.19 |

Table 3: Quantitative results of different methods on T2I-CompBench.

| Method | Non-Spatial | | | Complex | | | |
|---|---|---|---|---|---|---|---|
| | CLIP | B-CLIP | mG-C | CLIP | B-CLIP | 3-in-1 | mG-C |
| Stable Diffusion (Rombach et al., 2022) | 30.79 | 75.65 | 81.70 | 28.76 | 68.16 | 30.80 | 80.75 |
| GLIGEN (Li et al., 2023) | 31.56 | 76.97 | 82.41 | 31.14 | 69.54 | 33.41 | 81.22 |
| BoxDiff (Xie et al., 2023) | 31.12 | 76.30 | 81.91 | 30.06 | 68.89 | 32.84 | 80.94 |
| HiCo (Cheng et al., 2024) | 32.07 | 77.14 | **83.56** | 31.39 | 70.11 | 34.31 | 81.56 |
| DreamRenderer (Zhou et al., 2025) | 31.89 | 76.73 | 82.47 | 31.59 | 69.49 | 34.52 | 81.40 |
| Ours | **32.14** | **77.38** | 83.01 | **32.28** | **70.85** | **34.78** | **81.72** |

## 4.1 BENCHMARKS AND EVALUATION METRICS

**HICO-DET**(Chao et al., 2018) contains 47,776 images, with 38,118 for training and 9,658 for testing. In our experiments, we use the test set annotations as input to generate interaction images. To evaluate this, we employ the pre-trained HOI detector FGAHOI (Ma et al., 2023) to identify the HOI instances in the generated images and compare them with the ground truth from the original annotations in HICO-DET. We report the HOI Detection Score based on the FGAHOI protocol under the Default and Known Object.

**COCO-Position** (Lin et al., 2014) (800 sampled test images with captions as global prompts, category labels as instance descriptions, and original bounding boxes as layouts) evaluates spatial accuracy and image fidelity. We measure (1) mIoU and Grounding-DINO AP (Liu et al., 2024) for layout adherence; (2) FID (Heusel et al., 2017) for visual quality; and (3) CLIP (Radford et al., 2021) score for image–text consistency.

**T2I-CompBench** (Huang et al., 2023) (6000 compositional prompts: 1000 each for attribute binding, object relations and complex compositions; 300 per category for testing) challenges open-world relational generation. We generate layouts with LayoutGPT (Feng et al., 2023), then evaluate: (1) non-spatial relations via CLIP; (2) attribute binding via BLIP-CLIP ("B-CLIP") (Li et al., 2022); (3) multimodal reasoning via MiniGPT4-CoT (Zhu et al., 2023); and (4) spatial relations via UniDet (Zhou et al., 2022). For complex compositions, we follow the benchmark's 3-in-1 metric—averaging CLIP, B-VQA (disentangled BLIP-VQA) and UniDet.

## 4.2 QUANTITATIVE RESULTS

**HICO-DET.** Table 1 presents the test results of our proposed method compared to existing baselines on the HICO-DET. Compared to current baselines, our method achieves the best results. In terms of interaction accuracy, as a comparison, StableDiffusion only focuses on the semantic reconstruction. In contrast to other Layout-based models, our method introduces targeted improvements in modeling relations, leading to a significant performance boost in the reconstruction of object interactions.

**COCO-Position.** Table 2 presents the test results of our proposed method on the COCO-Position. The results demonstrate that our method has advantages in spatial accuracy. Furthermore, the FID score demonstrates that our improvements in spatial accuracy and text-image consistency are achieved without any perceptible degradation in image quality.

**T2I-CompBench.** As illustrated in table 3, Non-Spatial indicates the ability to reconstruct semantics during object interactions. Consistent with the results from the previous two benchmarks, our method continues to demonstrate an advantage in modeling object interactions. For complex scene generation, our proposed method achieves the best performance on 3-in-1, demonstrating that we retain strong

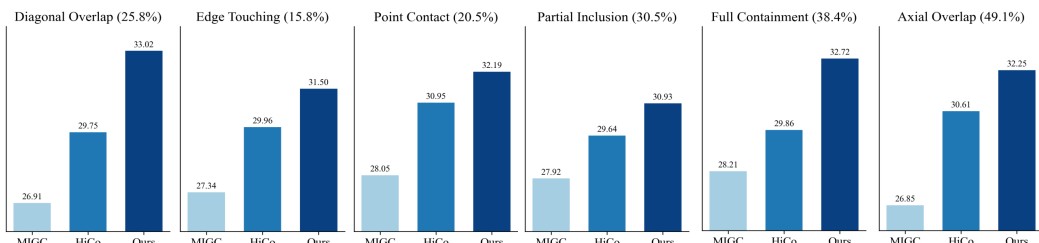

Figure 3: We conduct evaluations on the HICO-DET test set by statistically analyzing images containing the target relations (notably, some images may encompass multiple relations). The percentages denote the proportion of images containing each relation type relative to the entire test set. The HOI Detection Score based on the FGAHOI Swin-Tiny under the Default setting is reported.

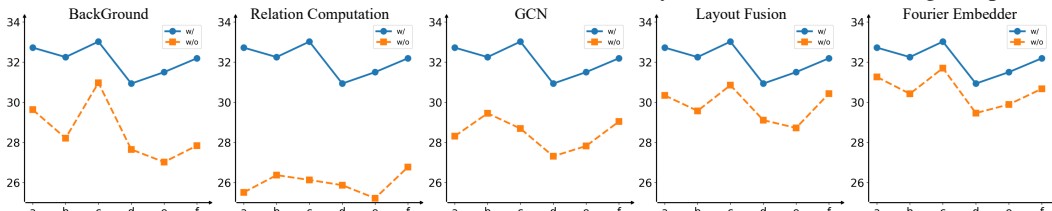

Figure 4: We conducted further ablation studies on the HICO-DET test set to evaluate the impact of various components, including background fusion and the modules detailed in Table 4. The results under different relation-dominated scenarios ((a)-(f)) are presented in the form of line charts. The HOI detection scores are based on the FGAHOI Swin-Tiny model under the default setting.

generative capabilities even when dealing with complex scenes.

**Generation Results for Individual Relations.** To examine whether generation effectiveness varies across relation types and how this impacts overall generation quality, we group images from the HICO-DET test set by specific relations and perform statistical analysis with baselines. The results, as shown in Fig. 3, demonstrate that our method consistently achieves preferable generation performance across all relation categories.

## 4.3 QUALITATIVE RESULTS

As shown in Fig. 5, we present the qualitative evaluation results of our approach. Our method demonstrates effective spatial control capabilities across diverse scenarios. First, compared to Stable Diffusion, our approach demonstrates a significantly higher accuracy in reconstructing the semantic content and relational semantics specified in image descriptions. For instance, in the 3rd example involving a complex scene, our method avoids issues such as missing human figures, which are observed in the results of Stable Diffusion. In comparison to other layout-based generation methods, it is evident that prior works fail to capture interactive actions such as "stand on" in Example 1, "grab" in Example 2. In contrast, our approach successfully generates these inter-object interactions.

## 4.4 ABLATION STUDY

We conduct ablation experiments on four key components: the use of Relation Bounding Boxes, the incorporation of object interaction information via GCN in processing Relation Boxes, the Layout Fusion module, and the Fourier Embedder. We evaluate the generated images on the HICO-DET dataset, and the evaluation results are shown in Table 4. valuation results on the HICO-DET dataset (Table 4) indicate that employing Relation Bounding Boxes enhances the model's ability to generate interactions between objects. The inclusion of GCN also leads to noticeable metric improvements, as it facilitates information exchange between objects and relations, thereby strengthening the representation of both. The introduction of the Layout Fusion module proves beneficial as well, demonstrating that integrating layout with image features helps better control object placement and reinforces the model's spatial constraint capability. Ablation results for the Fourier Embedder confirm its contribution to capturing spatial details. Illustrated in Fig. 4, we also evaluated the contribution of each module and background fusion to the generation of images dominated by various relationships.

Table 4: The ablation on the key components in our framework: RBC (the Relation Bounding Box Computation module), LF (the Layout Fusion module), and FE (Fourier Embedder).

| RBC | GCN | LF | FE | Non-Spatial | | Complex | |
|---|---|---|---|---|---|---|---|
| | | | | CLIP | B-CLIP | 3-in-1 | mG-C |
| ✓ | ✓ | ✓ | ✓ | **32.28** | **26.21** | **33.14** | **30.56** |
| | | ✓ | ✓ | 26.83 | 22.59 | 28.16 | 25.98 |
| ✓ | | ✓ | ✓ | 29.35 | 24.19 | 31.95 | 29.04 |
| ✓ | ✓ | | ✓ | 30.89 | 25.05 | 31.79 | 28.57 |
| ✓ | ✓ | ✓ | | 30.01 | 25.63 | 32.44 | 29.61 |

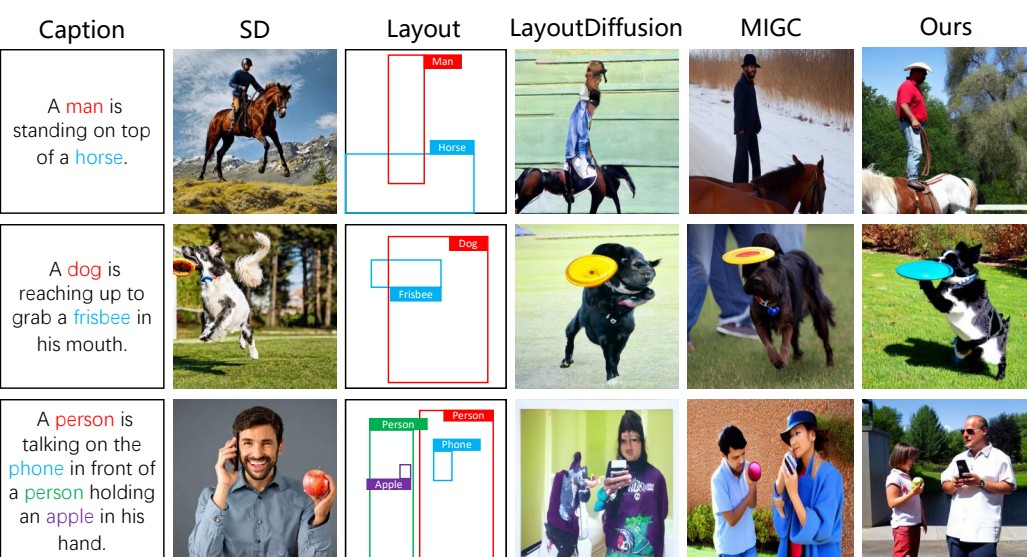

Figure 5: The qualitative results of our work are demonstrated in the figure above through a comparative analysis with layout-based approaches, where the generated images from Stable Diffusion serve as the baseline, thereby highlighting the generative outcomes achieved by our method.

Performance across all relation categories declined to varying degrees, indicating that these modules play a role in both enhancing and maintaining the consistency of various relations. Furthermore, the integration of background information also contributed to the faithful reproduction of objects in the generated images. We also observed that different relation categories exhibit varying sensitivities to certain modules. This suggests the potential for future work involving more granular, relation-specific module optimization or adaptive configuration based on distinct relational characteristics.

## 5    CONCLUSION

In this paper, we introduce Relation-Augmented Diffusion, a novel framework that advances layout-to-image generation by explicitly modeling inter-object relations and implicitly harmonizing background-object dependencies. By translating textual interactions into spatially grounded relation bounding boxes and leveraging GCN-based bidirectional reasoning, our method achieves precise control over both semantic coherence and spatial fidelity. Extensive experiments on several publicly available datasets demonstrate SOTA performance, with significant improvements in three benchmarks. While our method demonstrates strong performance in structured scene generation, several limitations and future directions warrant attention. For example, our emphasis on enhancing inter-object relations may be less effective in scenes involving a large number of interacting objects. Future research could extend this framework to video generation and 3D scenarios, further broadening its applicability.

ETHICAL STATEMENT

This research was conducted in compliance with ethical standards in artificial intelligence research. The study does not involve the collection of new human or animal data. All datasets used in our experiments are publicly available from the referenced publications or the links provided. The proposed models were designed and evaluated strictly for academic research purposes. Potential risks of misuse, bias, or unfairness in the model outputs were carefully considered. The authors declare no competing interests.

REPRODUCIBILITY STATEMENT

To ensure reproducibility, all datasets used in this work are publicly available and can be obtained from the referenced literature and websites. We provide a detailed description of our methodology and experimental settings in the Methods and Experiments sections and the code is also available.

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

## A    ADDITIONAL EXAMPLES

Figure 6: We generate more synthetic samples for the model.

## B    SPATIAL RELATION CLASSIFICATION IN RELATION BOUNDING BOX COMPUTATION

Let the bounding boxes of objects A and B be denoted as $\mathbf{b}_A = [x_1^A, y_1^A, x_2^A, y_2^A]$ and $\mathbf{b}_B = [x_1^B, y_1^B, x_2^B, y_2^B]$. We primarily consider two aspects: overlap and containment. The projection onto the x-axis and y-axis is denoted as $\mathbf{x}_p$ and $\mathbf{y}_p$. When there is an overlap between two objects on either one or both axes, it is necessary to determine whether an inclusion relationship exists along the overlapping axis. Based on this criterion, we classify the spatial relationship between two objects in a two-dimensional plane into the following six types:

(a) Full containment:    $x_{ol} \geq 0, \; y_{ol} \geq 0, \; \mathbf{b}_A \subseteq \mathbf{b}_B$

(b) Axial overlap:    $x_{ol} \geq 0, \; y_{ol} \geq 0, \; \mathbf{x}_p^A \subseteq \mathbf{x}_p^B$ or $\mathbf{y}_p^A \subseteq \mathbf{y}_p^B$

(c) Diagonal overlap:    $x_{ol} \geq 0, \; y_{ol} \geq 0, \;$ no axis exhibits containment.

(d) Partial inclusion:    $(x_{ol} \geq 0, y_{ol} < 0)$ or $(x_{ol} < 0, y_{ol} \geq 0), \; \mathbf{x}_p^A \subseteq \mathbf{x}_p^B$ or $\mathbf{y}_p^A \subseteq \mathbf{y}_p^B$

(e) Edge touching:    $(x_{ol} \geq 0, y_{ol} < 0)$ or $(x_{ol} < 0, y_{ol} \geq 0), \;$ no axis exhibits containment.

(f) Point contact:    $x_{ol} < 0, \; y_{ol} < 0.$

For example, for the diagonal overlap relation, the relation bounding box is computed as the intersection region between the two object bounding boxes:

$$x_1^r = \max(x_1^A, x_2^B), y_1^r = \max(y_1^A, y_1^B), x_2^r = \min(x_2^A, x_2^B), y_2^r = \min(y_2^A, y_2^B). \quad (11)$$

## C  THE USE OF LLMS

We used large language models in two limited, well-defined ways during manuscript preparation. First, LLMs were employed to aid and polish writing — improving clarity, grammar, and phrasing of sections and figure captions. All text suggestions produced by LLMs were reviewed, revised, and approved by the authors. Second, LLMs were used as a retrieval and discovery aid to help locate related work and summarize relevant papers during literature review. Outputs from retrieval tasks were treated as starting points: references and factual claims identified via LLM assistance were independently verified by the authors against primary sources.

