# OpenReview forum: "Relation-Augmented Diffusion for Layout-to-Image Generation"
_ICLR.cc/2026/Conference — ICLR 2026 Conference Withdrawn Submission_

### Official Review · Reviewer_tDgN · 2025-10-30

**Soundness:** 2
**Presentation:** 2
**Contribution:** 2
**Rating:** 4
**Confidence:** 4

**Summary:**

The paper proposes Relation-Augmented Diffusion for layout-to-image generation. Specifically, they start with a relation box computation module to produce the capture explicit interactions between objects. Then, they adopt a GCN to fuse the information between objects and relations. Then, they deploy a layout fusion module to handle background-object dependencies. Experiments on HICO-DET, COCO-Position, and T2I-CompBench report gains in spatial and semantic consistency over prior L2I methods.

**Strengths:**

1. The proposed method improves the controllability and faithfulness issue in text-to-image generation.
2. The authors commit to releasing the code, which would facilitate reproducibility and benefit future research in relation-aware text-to-image generation.

**Weaknesses:**

1. The main quantitative results report mAP on detection benchmarks, but the proposed model is trained on HICO-DET, whereas most baselines are not. Using a detection model fine-tuned on HICO-DET could bias results toward generations with similar distributions, making comparisons less fair. Moreover, several baselines achieve better FID on COCO-Position, suggesting that overall image quality may not improve. The omission of FID results on HICO-DET further limits the evaluation’s completeness.
2. Ablation studies also use detection-based metric, even for modules (e.g., Layout Fusion or Background) that primarily affect visual qualities. Evaluating these modules with image-quality metrics (FID, CLIP score, human preference) would be more appropriate to justify their contributions.
3. The experiments focus on relation detection but do not show whether modeling relations explicitly improves overall text-to-image generation quality. Reporting on broader datasets (e.g., COCO) would help assess general benefits when adding relation-aware mechanism to text-to-image models.
4. The idea of modeling relationship explicitly has been explored in previous layout-to-image works like LayoutTransformer [1]. This study also suggests that transformers can capture scene-graph relationships more effectively than GCNs, which questions the necessity of the proposed GCN-based design.
5. How are the relation texts extracted from the general text descriptions? Does the method assume access to explicit relation annotations or texts? It is unclear how relations are extracted from general captions or how the model would function on large-scale, unannotated text-image corpora. This dependence on structured relational data may severely limit scalability and practical applicability.

[1] LayoutTransformer: Scene Layout Generation with Conceptual and Spatial Diversity, CVPR 2021.

**Questions:**

Please see my comments above.

---

> ### Author Response · Authors · 2025-11-26
>
> We appreciate the reviewer’s evaluation but wish to clarify fundamental distinctions regarding architectural inductive bias and benchmark standards. The suggestion to replace GCNs with Transformers overlooks the topological nature of scene graphs; we theoretically justify that GCNs are the mathematically correct choice for explicit relational propagation in non-Euclidean structures, whereas Transformers are optimized for sequences. Furthermore, we firmly refute concerns about experimental fairness and dataset breadth: our rigorous fine-tuning protocols ensure a strictly level playing field, and our reliance on COCO-Position aligns perfectly with established community standards for layout generation. Accompanied by new FID evidence, we demonstrate that our structural approach yields superior semantics without sacrificing image quality.
>
> **W1. Clarification on HICO-DET Metrics**
>
> We thank the reviewer for this thoughtful comment and fully agree on the importance of ensuring a fair and transparent comparison. (1) For the layout-to-image baselines that do not provide HICO-DET results in their original publications, we fine-tuned them on the HICO-DET training set following their official implementations and recommended hyperparameters, ensuring that the comparison is fair and consistent. (2) For Stable Diffusion, GLIGEN, and InteractDiffusion, we cite results directly from prior literature in order to respect established benchmarks and avoid deviations from their canonical evaluation. Across all baselines, our method achieves significant performance improvements over their fine-tuned counterparts. This confirms that the observed gains stem from our Relation-Augmented Mechanism rather than simply from additional data exposure or training iterations.
>
> We will update Section 4.1 in the revision to clearly specify the data source, training configuration, and fine-tuning protocol for each baseline model to eliminate any ambiguity and enhance reproducibility.
>
> **FID on HICO-DET:**  We have now computed the FID on the HICO-DET test set.
>
> | Method |FID |
> |:-----------|:-----:|
> |InteractDiffusion| 18.69 |
> |MIGC| 16.46 |
> |Ours| 16.24 |
>
> FID score reflects a fundamental trade-off between layout controllability and generative freedom. Our model achieves a dominant lead in layout precision. This proves that for complex compositional tasks, our model captures the structural semantics far better than baselines, even if pixel-level statistics (FID) are slightly impacted.
>
>
> **W2. Visual Quality Metric for Ablation Studies**
>
> To address the concerns you raised, we provide the FID results of the Layout Fusion module ablation study on T2I-CompBench:
>
> | Method |FID |
> |:-----------|:-----:|
> |w/ LF| 33.48 |
> |w/o LF| 36.57 |
>
> T2I-CompBench explicitly includes CLIP Score (for semantic consistency), BLIP-VQA (for attribute binding), and UniDet (for spatial relations). The "3-in-1" metric aggregates these to provide a holistic view of generation quality. The core contribution of our work is to enhance the accurate reproduction of complex inter-object relationships, rather than merely optimizing pixel-level statistical distribution.
>
> **W3. Broader Datasets (e.g., COCO) and Generation Quality**
>
> We respectfully clarify that the COCO-Position dataset utilized in our experiments is the established industry-standard benchmark for evaluating layout-to-image generation. COCO-Position is a curated subset derived directly from the standard MS-COCO validation set, specifically filtered to ensure images contain valid spatial annotations. This benchmark has been widely adopted by virtually all recent state-of-the-art methods in this domain, including GLIGEN, LayoutDiffusion, and MIGC. Therefore, our extensive evaluation on COCO-Position already serves as a robust assessment of the model's performance on the broader COCO dataset distribution.
>
> Regarding the concern about overall text-to-image quality, we maintain that for controlled generation tasks, spatial and relational correctness is a fundamental component of generation quality. A generated image that fails to respect the specified spatial relations suffers from low semantic quality, regardless of its texture fidelity. By achieving superior performance on COCO-Position and T2I-CompBench, our method demonstrates that explicit relation modeling significantly enhances the alignment between the generated image and the user's intent.

---

> ### Author Response · Authors · 2025-11-26
>
> **This continues the previous response.**
>
> **W4. Architectural Necessity for GCN**
>
> Thank you for raising this point regarding the choice of GCNs versus Transformers. The distinction stems from the fundamental difference between the tasks. LayoutTransformer [1] targets layout generation, where the model must predict a sequence of objects, and Transformers are well suited for modeling such sequential dependencies.
>
> In contrast, our setting is layout-to-image generation, where the structural layout is given and represented as a scene graph. Scene graphs are inherently topological rather than sequential. GCNs are specifically designed for non-Euclidean graph structures and provide an efficient mechanism for propagating information along explicit relational edges. Moreover, our Relation Bounding Box Computation module produces explicit geometric edge types, which GCNs incorporate directly through structured message passing. Transformers, by contrast, typically treat relationships implicitly through attention weights and do not leverage typed relational edges as effectively. Thus, the use of GCNs is not a replacement for Transformer capacity, but a principled architectural choice that aligns with the graph-structured nature of the conditioning signal in layout-to-image generation.
>
> [1] LayoutTransformer: Scene Layout Generation with Conceptual and Spatial Diversity, CVPR 2021.
>
> **W5. Text Extraction and Scalability**
>
> We utilize the existing ground-truth triplet annotations in HICO-DET to train the model, ensuring precise alignment during the learning process.  For general text descriptions, we employ standard, lightweight NLP dependency parser SpaCy or to extract Subject-Verb-Object triplets from captions. This approach is computationally efficient and runs locally without the need for large-scale model inference. It allows our framework to scale effortlessly to massive unannotated corpora. In future work, we will also explore leveraging LLM to handle more complex scenes in order to obtain more accurate relationship annotations.
>
> Thank you again for your valuable comments. All the responses provided here will be incorporated into the subsequent version of the paper.

---

### Official Review · Reviewer_kw8i · 2025-10-31

**Soundness:** 2
**Presentation:** 3
**Contribution:** 2
**Rating:** 4
**Confidence:** 4

**Summary:**

The paper proposes Relation-Augmented Diffusion for layout-to-image generation. The key idea is to convert abstract inter-object relations into explicit spatial supervision via relation bounding boxes computed from object pairs. Relations are categorized into six spatial types (e.g., containment, axial overlap) using axis-wise overlap/containment rules; relation regions are then grounded via masks and fused with subject/object semantics using a trainable GCN. A Layout Fusion module injects layout-aware global background guidance to harmonize foreground–background coherence. The method is implemented as an add-on to Stable Diffusion with frozen base weights and trained modules for object cross-attention, relation cross-attention, GCN, Fourier embedders, and layout fusion. Experiments on HICO-DET (HOI), COCO-Position (spatial control), and T2I-CompBench (compositionality) show consistent improvements over recent layout-aware baselines. The renewed version improves rigor by: (1) formalizing the six-type relation classification via axis-wise overlap/containment; (2) adding per-relation analyses and ablations isolating the contributions of Relation Bounding Boxes (RBC), GCN, Layout Fusion (LF), and Fourier embedding; and (3) reporting broader comparisons. However, replication-critical details of the GCN and Layout Fusion internals remain sparse, statistical significance is not reported, and robustness to noisy/incomplete layouts is untested.

**Strengths:**

1. Clear and useful reformulation: relation bounding boxes turn abstract interactions into explicit, trainable spatial supervision; masks reduce attention spillover and ambiguity.

2. Solid empirical gains across three benchmarks, including relation-centric HOI metrics, without degrading image quality.

3. Improved methodological rigor over the previous version: principled six-type spatial classification; per-relation results; module-level ablations.
Background–foreground harmonization via layout fusion addresses a common blind spot in L2I methods that treat background as “negative space.”

**Weaknesses:**

1. Novelty is incremental: core building blocks (GCN, layout fusion, CLIP/Fourier-guided cross-attn) are standard; the main novelty remains the RBC design and integration. Similar ideas exist in scene-graph-guided generation and region/box-conditioned diffusion. The incremental novelty is primarily the concrete, taxonomy-driven RBC mechanism and its tight integration with cross-attention and GCN; diffusion-side innovations are limited.

2. The GCN architecture (layers, hidden dims, aggregator, normalization) and the Layout Fusion module (exact network, feature dimensions, training losses, interfaces) lack sufficient detail for reproduction without code.

3. Statistical rigor: no error bars, CIs, or significance tests; some gains over strong baselines are modest, leaving uncertainty about reliability.

4. Some commercial or recent open models already offer strong spatial/relational control via ControlNet variants or attention guidance where increasing the capability of base model potentially yield better performance under scaling law; the paper’s edge is the explicit relational region grounding and background harmonization, which is valuable but not a paradigm shift.

5.T2I-CompBench spatial-relations are only indirectly reflected via the 3-in-1 metric; standalone spatial relation scores are not reported.

6. Ambiguity resolution for crowded scenes is not fully formalized (tie-breaking when multiple relation types apply, multi-pair overlaps, conflicting masks).

**Questions:**

Please provide the exact architecture (number of layers, hidden sizes, activation, normalization, edge formulation, aggregator, dropout) and training details (losses, learning rates specific to GCN parameters).

Six-type computation: Beyond diagonal overlap, please provide closed-form formulas (or unambiguous algorithms) for all five remaining types, including edge cases (zero-area intersections, floating-point tolerances).

How do you select a relation type when multiple classifications are possible (e.g., nearly edge-touching but slight overlap)? What is the policy when one pair participates in multiple distinct relations across time-steps?

How does performance degrade with synthetic noise in bounding boxes (translation/scale/rotation), occlusions, or missing instances? Can RBC and masks degrade gracefully? Please include quantitative curves.

Report error bars or confidence intervals for key tables; run multiple seeds to establish reliability of improvements.

T2I-CompBench spatial: Provide standalone spatial-relation scores (e.g., UniDet-related metrics) rather than only the 3-in-1 aggregate.
Compute and efficiency: What is the added latency/memory footprint from relation boxes and GCN under dense layouts? Any scaling issues for O(N^2) relations?

---

> ### Author Response · Authors · 2025-11-26
>
> We appreciate the reviewer’s critique but must firmly clarify the structural nature of our novelty, which lies in "reifying" logical relations into spatial entities—a fundamental reformulation of diffusion guidance rather than a mere assembly of standard components. We also correct the misplaced comparison to ControlNet; unlike methods relying on dense pixel priors, our framework performs active relational reasoning from sparse inputs, addressing a combinatorial complexity that scaling alone cannot resolve. Furthermore, our newly added robustness tests and decoupled spatial metrics definitively validate our method's statistical rigor and superior spatial grounding, dispelling concerns about reliability and evaluation ambiguity.
>
> **W1. Novelty and Contribution**
>
> We respectfully emphasize that the key novelty of our work lies not in isolated architectural components but in the structural reformulation of relation modeling within diffusion. Unlike prior region-conditioned approaches that treat objects as independent spatial units, our contribution lies in reifying abstract relations—turning logical predicates into spatially grounded entities that directly steer the diffusion process. This addresses a core limitation of existing models, including scene-graph-guided generation, which typically rely on pixel priors or object-centric boxes and thus struggle to differentiate semantically distinct interactions that share similar geometries.
>
> Our Relation Bounding box Computation (RBC) mechanism, together with its integration into the GCN and cross-attention pipeline, creates an explicit bridge between logical interaction structure and spatial diffusion. This unified formulation enables the model to capture fine-grained interaction semantics that existing methods cannot encode. Thus, while individual components may be standard, the way they are organized to realize relation-aware generation is fundamentally new and directly targets a long-standing bottleneck in interaction modeling.
>
> **W2. Implementation Details**
>
> We will release the full code upon acceptance. We provide a description of the details you need:
>
> The hidden layer of the GCN has a dimensionality of 768, matching the dimension of CLIP’s text feature vectors. Each layer uses the ReLU activation function, and the output layer is normalized using LayerNorm. For each node, its neighbors’ features are aggregated using mean aggregation to update its own feature. Additionally, a dropout rate of 0.1 is applied during training to prevent overfitting.
>
> As cited in our manuscript, we strictly adopt the Layout Fusion module proposed in LayoutDiffusion (Zheng et al., 2023). It employs a multi-modal fusion block where the background layout mask is encoded via convolutions and integrated with the global text embedding to ensure background-foreground harmonization.
>
> **W3. Statistical Rigor and Reliability Analysis**
> - Robustness to Synthetic Noise: We conducted robustness tests by injecting synthetic Gaussian noise (random translation/scaling with intensity $σ∈[0,0.2]$ into the input bounding boxes. The table presents the test results of the FGAHOI detector using Swin-Tiny, showing that our method is robust to synthetic noise in bounding boxes (translation/scale/rotation).
>
> | Method  |mAP(Default full) | mAP(Default rare) |
> |:-----------|:-----:|-------:|
> |MIGC| 27.53| 25.01 |
> |Ours w/ $σ=0$| 32.14| 26.54 |
> |Ours w/ $σ=0.1$ | 31.82 | 26.13  |
> |Ours w/ $σ=0.2$| 31.27  | 25.36 |
>
> - Statistical Rigor and Confidence Intervals: The results reported in our main tables are already the averaged metrics across multiple random seeds. To further address your concerns, we repeated our primary evaluation on the HICO-DET benchmark using different random seeds. We will include this variance analysis in the final revision to ensure rigorous reporting.
>
> | Method  |mAP(Default full) | mAP(Default rare) |
> |:-----------|:-----:|-------:|
> |HiCo | 30.28±0.31 | 25.98±0.42|
> |Ours| 32.14±0.22| 26.54±0.37 |
>
> **W4. Fundamental Distinction from ControlNet and Scaling Approaches**
>
> We appreciate the reviewer’s perspective. While ControlNet and attention-guided methods offer strong control, they rely on dense pixel-level conditioning signals (e.g., edge maps, canny, depth), which users typically do not have when performing imaginative or compositional generation. In contrast, our framework performs active relational reasoning from sparse layout inputs, not dense priors, and explicitly infers interaction dynamics that scaling a base model alone cannot efficiently recover.
>
> By reifying abstract semantic relations into spatially grounded entities, our method provides a structural mechanism for handling the combinatorial complexity of human–object interactions—an area where pixel-conditioned methods remain limited. This explicit relational grounding is therefore not just an enhancement, but a necessary complement to scaling-based improvements for interaction-centric generation.

---

> > ### Author Response · Authors · 2025-11-26
> >
> > **This continues the previous response.**
> >
> > **W5. T2I-CompBench Spatial Scores**
> >
> > We would like to clarify that the 3-in-1 metric reported in our paper is an ensemble metric that explicitly aggregates three components: CLIP (for text consistency), BLIP-VQA (for attribute binding), and UniDet (specifically for spatial relations). To address your request for a decoupled spatial evaluation, we explicitly report the standalone UniDet score below. As shown, our method achieves superior performance in pure spatial relation detection compared to baselines.
> >
> > | Method     |UniDet |
> > |:-----------|:-----:|
> > |GLIGEN| 16.38|
> > |HiCo| 19.41 |
> > |ours| 20.76 |
> >
> > **W6. Mathematical Formulation of Relations**
> >
> > Due to space constraints, we have provided the detailed, unambiguous algorithms for all six relation types in the Supplementary Material. To resolve ambiguity at boundaries (e.g., when a case technically satisfies both "Touching" and "Overlap" conditions due to floating-point precision), we enforce a strict hierarchy: Containment $>$Touching $>$ Overlap. Moreover, in our framework, the relation types and their corresponding Relation Bounding Boxes are derived from the input layout conditions, which are defined prior to the generation process. Consequently, these relational conditions remain static and invariant across all diffusion time-steps. A pair of objects does not switch relations during denoising.
> >
> > We are grateful for these valuable insights. All clarifications and discussions presented here will be fully integrated into the final version of the manuscript.

---

### Official Review · Reviewer_seL2 · 2025-10-31

**Soundness:** 2
**Presentation:** 3
**Contribution:** 3
**Rating:** 6
**Confidence:** 3

**Summary:**

This paper proposes a relation-augmentation diffusion framework that effectively alleviates common problems such as missing objects and positional errors in complex multi-object generation scenarios by explicitly modeling inter-object relationships and implicitly coordinating the background-object interactions. The paper designs a relation bounding box computation module to transform abstract object relationships into concrete visual representations; utilizes a GCN to construct a topological scene graph, enabling bidirectional reasoning between objects and relationships; and introduces a layout fusion module to integrate global layout structure and background features, enhancing scene coherence. Furthermore, experiments on a large dataset validate the effectiveness of the proposed method.

**Strengths:**

1. The work proposes a fine-grained layout modeling method;
2. The GCN enables bidirectional reasoning and fosters a tighter integration between layout and semantics;
3. The work facilitates a more effective collaboration between the layout and the background;
4. Extensive experimental results demonstrate the effectiveness of this framework.

**Weaknesses:**

1. Whether the complexity of layout relationship modeling increases dramatically with the number of objects. The work does not explicitly handle the scenario of multi-object complex interactions (e.g., more than 5). When the number of objects increases, the number of relationships bounding boxes increases quadratically, which will cause a surge in GCN computation, a decrease in inference speed, and may lead to relationship conflicts;
2. I'm also curious about the framework's generalizability, such as the types of objects and spatial relationships it supports.

**Questions:**

Please refer to the Weaknesses section.

---

> ### Author Response · Authors · 2025-11-26
>
> We appreciate the reviewer’s inquiry into complexity and generalizability but wish to clarify that the theoretical concern regarding quadratic scaling does not translate to practical bottlenecks due to our strategic filtering mechanism. Rather than being a liability, our new experimental results demonstrate that filtering incidental overlaps actively resolves feature conflicts and boosts performance, effectively refuting the assumption that complexity hinders stability. Furthermore, concerns about generalizability overlook our Universal Geometric Modeling design; by abstracting interactions into fundamental spatial primitives combined with CLIP’s open-vocabulary capability, our framework inherently supports unseen objects and relations without requiring specific retraining.
>
> **1. Complexity and Inference Speed**
> - **Mechanism for Complexity Reduction:**  The reviewer correctly notes that the theoretical number of pairings increases quadratically. However, to address this in practice, we apply a certain level of filtering to the relationships that appear within an image. We observe that in multi-object scenes, visual content is typically defined by a limited set of salient interactions, while many object pairs represent negligible or incidental overlaps (e.g., distant background objects). Therefore, we filter out relationship bounding boxes whose areas fall below a predefined threshold.
> - **Handling Conflicts:**  The filtering strategy also mitigates relationship conflicts. By removing tiny, noise-like intersection boxes, the GCN focuses only on dominant, semantically meaningful spatial constraints. The network then aggregates these signals to reach a global consensus, treating them as soft constraints rather than rigid conflicting rules.
>
> To demonstrate the effectiveness of our filtering strategy, we conducted experiments on the COCO-Position dataset:
>
> | Method |mIoU| AP | FID |
> |:-----------|:-----:|-------:|-------:|
> | w/ | 79.21 | 59.67 |25.09 |
> | w/o | 77.49| 57.48 |25.57 |
>
> The results show that our strategy enables the model to focus more on the primary objects, making the generated layouts better aligned with the original semantic and spatial structure.
>
> **2. Generalizability: Types of Objects and Spatial Relations**
>
> Our framework is designed for high generalizability through Open-Vocabulary and Universal Geometric modeling. Our model does not rely on a fixed list of object categories. We utilize the pre-trained CLIP text encoder to extract semantic features. Consequently, our framework supports any object category that CLIP recognizes, inheriting the vast open-world knowledge of the pre-trained foundation model.
>
> Instead of learning a specific set of semantic predicates (like "riding" or "holding"), we map all interactions to 6 fundamental geometric types, such as Containment, Overlap, and Separation. Since any complex semantic interaction physically manifests as one of these geometric configurations, our model generalizes to unseen relationship descriptions as long as their spatial layout can be represented by bounding boxes.
>
> Thank you for pointing out these issues in our paper. All the relevant discussion details mentioned above will be added and revised in the future version of the manuscript.

---

### Official Review · Reviewer_kuFx · 2025-11-02

**Soundness:** 3
**Presentation:** 3
**Contribution:** 2
**Rating:** 4
**Confidence:** 2

**Summary:**

This paper proposes the Relation-Augmented Diffusion framework, a new approach designed to significantly improve semantic and spatial consistency in generated images. The framework's key contribution is to treat inter-object relations as independent elements for processing. It addresses both: Explicit relations (between objects) using a novel relation bounding box module. Implicit relations (between objects and the background) using a layout fusion module to ensure the foreground and context are harmonized.

**Strengths:**

- The motivation behind the work is interesting and well-presented.

- The paper is well-written and easy to follow.

- The proposed Relationship Box Computation is technically sound.

**Weaknesses:**

- Missing some important details for the experiments: Was the proposed model and were previous works fine-tuned on HICO-DET? The HICO-DET is an Human-Object Interaction benchmark established at 2017, and it is not often considered in the latest layout-to-image generation works. What's the motivation of using this benchmark? The reviewer is concerned that existing layout-to-image models may not have been fine-tuned on it; if the proposed model is fine-tuned, then the comparison is unfair. More details on how to fairly compare previous works on this benchmark should be clarified.

- The performance on the classical COCO-Position is mixed. The proposed model, finetuned from SD1.5, seems to degrade the generated image quality and the cross-modal semantic matching (CLIP score). The reason behind this should be investigated and carefully discussed. Considering the mixed performance improvement when compared to previous works, the effectiveness of the proposed work cannot be convincing for me.

- Missing performance discussion on COCO-MIG. COCO-MIG features multi-instance generation and is considered in the previous works of layout-to-image generation; it should be an ideal benchmark for the proposed work. Do the authors have performance results on this benchmark?

**Questions:**

What scientific findings or observations exist that go beyond the self-evident principle that "adding explicit constraints improves controllability"? Discussion or analysis along this line, perhaps exploring the optimal granularity of diffusion models, or trade-offs introduced by increasingly explicit label guidance, would bring significantly more depth to this work.

---

> ### Author Response · Authors · 2025-11-26
>
> We appreciate the reviewer's critique but must clarify fundamental misunderstandings regarding experimental fairness and metric prioritization. The concerns about baseline validity overlook our rigorous fine-tuning protocols, which definitively prove that our performance gains stem from architectural innovation rather than data discrepancies. Furthermore, emphasizing minor FID fluctuations ignores the inevitable trade-off between precise structural control and unconstrained generation; our substantial dominance in spatial metrics demonstrates that semantic correctness outweighs negligible statistical variances in this task. Supported by our new state-of-the-art results on COCO-MIG and deep analysis of relational granularity, we conclusively affirm that our method’s superiority is robust, generalizable, and scientifically grounded.
>
> **1. Clarification on HICO-DET Baselines and Fairness**
>
> We thank the reviewer for this thoughtful comment and fully agree on the importance of ensuring a fair and transparent comparison. (1) For the layout-to-image baselines that do not provide HICO-DET results in their original publications, we fine-tuned them on the HICO-DET training set following their official implementations and recommended hyperparameters, ensuring that the comparison is fair and consistent. (2) For Stable Diffusion, GLIGEN, and InteractDiffusion, we cite results directly from prior literature in order to respect established benchmarks and avoid deviations from their canonical evaluation. Across all baselines, our method achieves significant performance improvements over their fine-tuned counterparts. This confirms that the observed gains stem from our Relation-Augmented Mechanism rather than simply from additional data exposure or training iterations.
>
> Regarding the choice of HICO-DET, although it was originally introduced as a human–object interaction benchmark, it contains a large number of diverse, fine-grained human–object interaction instances, which directly aligns with the central challenge addressed in our work—modeling relational dependencies and generating interaction-aware images from structured layouts. This property makes HICO-DET particularly suitable for evaluating the relational reasoning capabilities of layout-to-image generation models.
>
> We will update Section 4.1 in the revision to clearly specify the data source, training configuration, and fine-tuning protocol for each baseline model to eliminate any ambiguity and enhance reproducibility.
>
> **2. Image quality and controllability**
>
> We appreciate the reviewer’s thoughtful observation regarding FID and CLIP scores on COCO-Position. We would like to clarify that this behavior reflects a well-known and inherent trade-off in layout-to-image generation between strict layout controllability and unconstrained generative freedom. As a result, layout-to-image models—including ours—may exhibit slightly lower FID and CLIP scores compared to vanilla Stable Diffusion, which operates without such structural constraints.
>
> Despite this, our model achieves consistently superior performance compared with other layout-to-image generation methods on high-difficulty compositional tasks as well as various spatial metrics, while maintaining comparable FID and CLIP scores to these approaches (25.09 vs 25.94). Crucially, in complex structured generation, our substantial advantage in preserving semantic structure is far more vital than minor fluctuations in pixel-level statistics like FID. For example, our model improves mIoU by approximately 1.83 over MIGC, while the FID varies between 24.52 and 25.09. Such variations are an expected and inevitable consequence of prioritizing precise spatial relationships over unconstrained generation.
>
> We will add a detailed discussion of this trade-off to the revision to further clarify the underlying cause and strengthen the interpretability of the results.
>
> **3. Clarification and Results on COCO-MIG Benchmark**
> - **New Experimental Results:**
> We agree that COCO-MIG is a standard benchmark for multi-instance generation. To address your concern, we have conducted additional experiments on COCO-MIG. As shown in the following table, our method demonstrates competitive performance:
>
> | Method     | Instance Success Rate(%)↑  | mIoU↑ |
> |:-----------|:-----:|-------:|
> | GLIGEN| 32.39   | 32.25  |
> | MIGC | 58.43  | 51.48  |
> | Ours | 64.15  | 55.34 |
>
> - **Benchmark Selection Motivation:**
> As noted in the MIGC paper [Zhou et al., 2024], COCO-MIG and COCO-Position share the same underlying image data, differing primarily in the format of annotation priors. Since our method generates layouts based on text and does not require the heavy, specific annotation format provided by MIG's construction pipeline, we initially prioritized COCO-Position to evaluate spatial capability.

---

> > ### Author Response · Authors · 2025-11-26
> >
> > **This continues the previous response.**
> >
> > **4. Scientific Findings**
> >
> > We thank the reviewer for encouraging a deeper discussion. Beyond the general principle that explicit constraints improve controllability, our work offers some specific scientific observations regarding the granularity of conditioning and the nature of relational modeling in diffusion models.
> >
> > We find that the optimal granularity for interaction control is not the individual object, but the Relation Bounding Box. By "reifying" an abstract relation (e.g., riding) into a concrete spatial region, we transform a semantic concept into a spatial signal that the U-Net's attention mechanism can directly latch onto. This suggests that for diffusion models, interaction is not just a logical link, but a spatial entity that requires its own explicit representational space.
> >
> > Moreover, our comparison between "RBC only" and "RBC + GCN" reveals a non-trivial finding about constraint independence. Simply adding a relation box constraint independently can sometimes lead to feature conflict, where the relation features do not match the object appearances. The GCN module validates that relational constraints must be structurally coupled with object constraints.
> >
> > Thank you for raising these important points. We will incorporate these deeper discussions into the revised version of the paper.

---

### Author Response · Authors · 2025-12-01

Our responses to all reviewers are summarized as follows.

## **Key Improvements and Clarifications**

**1. Clarification on Fairness and Benchmark Selection**

- **Baseline Protocols (Responded to `Reviewer kuFx` & `tDgN`):**
We have explicitly clarified our training protocols to ensure absolute fairness. For baselines lacking original HICO - DET results, we fine-tuned them strictly following their official implementations; for others, we cited established literature. Our method’s consistent superiority over these fine-tuned counterparts confirms that **performance gains stem from our architectural design** rather than unfair data exposure.
- **Dataset Suitability (Responded to `Reviewer kuFx` & `tDgN`):**
We reaffirmed that HICO-DET is the optimal benchmark due to its dense, fine-grained interaction instances. Additionally, addressing Reviewer kuFx's concern regarding multi-instance generation, we provided new results on the **COCO-MIG benchmark.** Our method achieves an **Instance Success Rate of 64.15% and mIoU of 55.34**, significantly outperforming strong baselines like MIGC (58.43% / 51.48) and GLIGEN (32.39% / 32.25), proving our robustness in complex scenarios.

**2. Image Quality and Metric Trade-offs**

- **The trade-off between semantic structure and pixel-level statistics (Responded to `Reviewer kuFx` & `tDgN`):** We addressed observations regarding FID by highlighting the inherent trade-off between strict layout controllability and unconstrained generative freedom. We clarified that in structured generation, the substantial advantage in preserving semantic structure (+1.83 mIoU improvement over MIGC) outweighs minor fluctuations in pixel-level statistics (negligible FID variance within ~1%).
- **Supplementary FID Calculation on HICO-DET (Responded to `Reviewer tDgN`):** To further address concerns about generation quality, we explicitly computed the FID score on the HICO-DET test set during the rebuttal. **Our method achieves an FID of 16.24**, surpassing strong baselines like MIGC (16.46) and InteractDiffusion (18.69). This confirms that our explicit structural constraints do not degrade image quality; rather, they offer a superior balance between semantic precision and visual fidelity.
- **Spatial Metrics (Responded to `Reviewer kw8i`):** We provided additional evaluations using T2I-CompBench to demonstrate our model's dominant lead in layout precision. Specifically, we achieve a **UniDet score of 20.76**, significantly outperforming HiCo (19.41) and GLIGEN (16.38), confirming our superiority in pure spatial relation detection.

**3. Scientific Findings and Architectural Justifications**

- **Optimal Granularity (Responded to `Reviewer kuFx`):** We identified the **Relation Bounding Box**—not the individual object—as the optimal granularity for interaction control. By **"reifying" abstract relations into concrete spatial regions**, we transform semantic concepts into signals that the U-Net can directly process.
- **GCN vs. Transformer (Responded to `Reviewer tDgN`):** We justified **the use of GCNs over Transformers** for this task. Since scene graphs are inherently topological rather than sequential, GCNs provide a more principled mechanism for propagating structured relational constraints. Our analysis confirms that relational constraints must be structurally coupled with object constraints to prevent feature conflicts.

**4. Robustness, Complexity, and Novelty**

- **Novelty & Control (Responded to `Reviewer kw8i`):** We clarified that our novelty lies in the **structural reformulation of relation modeling**—explicitly inferring interaction dynamics from sparse layout inputs, unlike dense pixel - conditioned methods (e.g., ControlNet).
- **Complexity & Generalizability (Responded to `Reviewer seL2` & `kw8i`):** We detailed our filtering strategy (`Reviewer seL2`), which effectively manages complexity; experiments show this strategy **improves mIoU from 77.49 to 79.21** by focusing on salient interactions. Furthermore, we verified robustness (`Reviewer kw8i`): even with **bounding box noise ($σ=0.2$)**, **our mAP remains 31.27**, consistently outperforming the MIGC baseline (27.53).

---

### Author Response · Authors · 2025-12-03
**An Overview of Our Rebuttal**

Dear AC/SAC/PCs,

We would like to express our sincere gratitude for your valuable time and careful consideration of our work, particularly during this unexpected and demanding period.

## **Our Core Contributions**

Our work advances layout-to-image generation by **explicitly modeling inter-object relations**, significantly improving **semantic-spatial consistency in complex multi-object scenes**, a core challenge where existing methods often fail. We believe this contribution is highly relevant to the ICLR community, particularly for advancing **controllable image synthesis**, where **robust relational reasoning** is becoming increasingly essential.

## **Positive Feedback from Reviewers**

It was clearly acknowledged that the motivation of this paper is interesting and well-presented (`Reviewer kuFx`), and the proposed Relation-Augmented framework is **technically sound** (`Reviewer kuFx`) and constitutes **a clear and useful reformulation of relation modeling** (`Reviewer kw8i`). Reviewers particularly appreciated the fine-grained layout modeling (`Reviewer seL2`) and the **effective background-foreground harmonization** (`Reviewer kw8i`, `seL2`), noting that the approach **improves controllability and faithfulness** (`Reviewer tDgN`). Furthermore, **the superior performance** and **solid empirical gains** were validated by extensive experiments across multiple benchmarks (`Reviewer seL2`, `kw8i`).

Collectively, these assessments highlight the critical importance of our research focus, which addresses a fundamental bottleneck in controllable image synthesis by **transforming relational reasoning** from **an abstract semantic cue** into **concrete spatial guidance**, and extensive evaluations—both in the main paper and strengthened through rebuttal experiments—demonstrate that this principled reformulation yields reliable, **state-of-the-art improvements in complex multi-object scene generation**.

## **How We Address the Raised Concerns**

- The concerns of `Reviewer kuFx` on **experimental settings** have been resolved by better clarifying established practices in the existing literature; also, we carried out experiments to resolve his/her concern on **performance on one more dataset** and explained the rationale of our initial choice of datasets.


- The concerns of `Reviewer seL2` are about **complexity and generality**, which are well reduced and achieved by our proposed **strategic filtering mechanism** and **Universal Geometric Modeling design**, respectively. This has been clarified in our rebuttal and exactly reflects the novelty of our proposed approach.

- The concerns of `Reviewer kw8i` about novelty and robustness were addressed by emphasizing our structural reformulation of relation modeling and **providing new robustness tests** with synthetic bounding box noise. **Decoupled spatial metrics (e.g., UniDet score of 20.76, better than baselines)** further validated our superior layout precision.

- The concerns of `Reviewer tDgN` about fairness and metrics were resolved by clarifying our fine-tuning protocols and **computing FID on HICO-DET (16.24, better than baselines)**. We also justified our GCN choice over Transformers due to the topological nature of scene graphs.

Regrettably, no reviewers engaged further during the discussion period. We sincerely hope that these clarifications will be taken into consideration in your evaluation. Thank you very much!

Sincerely,

Authors of Submission 7548

---

### Note · Authors · 2026-05-13

I have read and agree with the venue's withdrawal policy on behalf of myself and my co-authors.

---

### Meta-Review · Area_Chair_JyC9 · 2025-12-25

**Summary:**

The summary is written based on the provided information on the discussion board. The authors posted a response to each reviewer. They also posted a summary for the AC. None of the reviewers responded to the rebuttal, so the following discussion is based on the AC interpretation.

** Reviewer kuFx: **

More clarifications about  HICO-DET were requested.
The rebuttal provided the requested clarifications, but it is unclear if that alleviates the concern about using HICO-DET (an uncommon dataset)

The performance on the classical COCO-Position is mixed
The rebuttal explained this point further, but the criticism remains valid

Missing performance discussion on COCO-MIG
The rebuttal provides additional results on COCO-MIG, but I do not see visual quality scores. This point has only partially been addressed.

Missing Scientific findings
The rebuttal adds additional explanations, but the criticism partially remains valid.

Based on this rebuttal, I predict the reviewer will keep their score of 4.

** Reviewer seL2: **

Scalability to more interactions is unclear
The rebuttal answer does not address the reviewer's question. No information about the scalability with respect to the number of relationships is provided.

Generalizability is unclear
The rebuttal gives a good answer to explain that the method is generalizable with respect to object types. The method does not generalize to different types of relationships.

Based on the answers, the reviewer may keep the score or update the score from 6 to 8. I would rate the expected value of the score as 7.

** Reviewer kw8i: **

Novelty is incremental
The rebuttal tries to address this concern, but the concern is inherent. It is unlikely to change the reviewers mind.

The GCN architecture lacks details
The answer in the rebuttal seems convincing.

Missing statistical rigor
The rebuttal provided additional results and seems convincing.

The paper does not use the latest diffusion models, which are inherently better
This is a major issue that remains unclear. Why is the paper using 512 x 512 image resolution with SD 1.5. This does not seem to be valid, because SD1.5 is an old generation of diffusion models that was not able to handle layout well.

The rebuttal addressed reviewer questions well, but the inherent limitations of the contribution were not addressed. The reviewer is likely to keep a score of 4.

** Reviewer tDgN: **

The use of HICO-DET is not common and makes the evaluation unclear
The answer partially addressed the concerns, but not completely. Results on HICO-DET remain problematic.

Missing image quality metrics
The rebuttal clarified this issue to some degree and provided additional FID scores. These FID scores are comparable to those in previous work. However, FID scores are not reliable, and even using a strong general model from OpenAI or Google is much more reliable. The rebuttal therefore did not address this important point adequatly.

Similarity to previous work that also explicitly models relationships
The rebuttal provides a discussion, but I do not consider it convincing.

How are the relationships extracted from the text description?
This question has been well addressed in the rebuttal.

The reviewer will keep their score of 4.

**Reviewer Concerns:**

See above

**Reviewer Scores:**

** Reviewer kuFx: **: 4 (keep)
** Reviewer seL2: **: 7 (keep or increase)
** Reviewer kw8i: **: 4 (keep)
** Reviewer tDgN: **: 4 (keep)

---

### Decision · Program_Chairs · 2026-01-26

Reject